# Speeding-up while growing-up: Synchronous functional development of motor and non-motor processes across childhood and adolescence

**Kamila Śmigasiewicz**[1]*, **Mathieu Servant**[2], **Solène Ambrosi**[3], **Agnès Blaye**[3‡], **Borís Burle**[1‡]*

**1** Laboratoire de Neurosciences Cognitives, Aix-Marseille Université, CNRS, Marseille, France,
**2** Laboratoire de Recherches Intégratives en Neurosciences et Psychologie Cognitive, Université de Franche-Comté, Besançon, France, **3** Laboratoire de Psychologie Cognitive, Aix-Marseille Université, CNRS, Marseille, France

‡ These authors are joint senior authors on this work.
* k.smigasiewicz@gmail.com (KS); boris.burle@univ-amu.fr (BB)

## Abstract

Describing the maturation of information processing in children is fundamental for developmental science. Although non-linear changes in reaction times have been well-documented, direct measurement of the development of the different processing components is lacking. In this study, electromyography was used to quantify the maturation of premotor and motor processes on a sample of 114 children (6–14 years-old) and 15 adults. Using a model-based approach, we show that the development of these two components is well-described by an exponential decrease in duration, with the decay rate being equal for the two components. These findings provide the first unbiased evidence in favour of the common developmental rate of nonmotor and motor processes by directly confronting rates of development of different processing components within the same task. This common developmental rate contrasts with the differential physical maturation of region-specific cerebral gray and white matter. Tentative paths of interpretation are proposed in the discussion.

## Introduction

From childhood to adulthood, cognitive functions (defined broadly as mental processes underlying perception, attention, memory, reasoning, decision making, problem solving, action selection and control, etc. . .) undergo massive non-linear changes: they develop considerably during early and middle childhood and more slowly in later childhood and early adolescence, which is obvious in response time (RT) studies. RTs decrease sharply until age of 9–10 years, and more gradually beyond that age. Such a non-linear decrease (well described by a negative exponential function–see below for details) is ubiquitous and has been found to be true for a large variety of RT tasks [1]. RT, however, is an aggregated measure of cognitive processes that reveals the final output of a series of complex information processing

**Citation:** Śmigasiewicz K, Servant M, Ambrosi S, Blaye A, Burle B (2021) Speeding-up while growing-up: Synchronous functional development of motor and non-motor processes across childhood and adolescence. PLoS ONE 16(9): e0255892. https://doi.org/10.1371/journal.pone.0255892

**Data Availability Statement:** Data and scripts for all analysis can be found under the Open Science

Foundation link of this project: https://osf.io/
93rmd/ (DOI: 10.17605/OSF.IO/93RMD).

**Funding:** This work was supported by French
Agence Nationale de la Recherche (ANR-15-CE28-
0008, DOPCONTROL) awarded to A.B and B.B.
https://anr.fr. The funders had no role in study
design, data collection and analysis, decision to
publish, or preparation of the manuscript.

**Competing interests:** The authors have declared
that no competing interests exist.

transformations (at a minimum, perception, decision, and response execution). Characterizing the developmental pattern of these different processes is a fundamental issue. The objective of the present work is to study whether a non-linear evolution characterizes the maturation of the different processes underlying RT and, if so, do they have similar dynamics or do they develop very differently.

Previous attempts aiming at delimiting different processes aggregated in RT measure relied on comparing tasks or task conditions [1, 2] associated to experimental manipulations (for example: simple- and choice RT tasks, visual search, mental rotation, same-different judgement tasks, Stroop task or classification task) thought to affect specific component processes. Remarkably, not only the changes of RT durations with age measured in a large variety of different cognitive task conditions were always non-linear, but also the trajectories were very similar. Therefore, it has been proposed that all cognitive processes develop at the same rate [1]. However, subtracting the RT of one condition from the other to draw inference on the assumed manipulated process (often termed "subtractive method"), requires additional assumptions. First, one has to assume "pure insertion" of processes, that is, a mental process can be added or omitted from the processing chain without changing the duration of other processes [3]. In the development literature, it must further be assumed that 1) the nature of information processes engaged in a task are identical for all ages [4] and 2) the processes inserted are the same for all ages. However, the assumption about pure insertion of processes is not necessarily valid [see 4, 5 for criticisms], and it is inconsistent with continuous flow models of information processing, for which empirical evidence has been provided [e.g., 6, 7].

To address these shortcomings, Ridderinkhof and van der Molen [8] used event-related potentials (ERPs) of EEG to estimate chronometric properties of mental processes operating within RT [see also 9]. ERPs are series of positive and negative deflections of brain electrical activity that are time-locked to stimulus presentation or response execution. As such, they index stimulus- and response-related cognitive operations. Ridderinkhof and van der Molen [8] examined the latencies of P3 component (assumed to be related to stimulus processing) and the lateralized readiness potentials (LRP; assumed to be related to motor preparation). They observed that the latencies of both potentials evolve across childhood in a non-linear way. However, the P3 component developed faster than the LRP component, which led these authors to conclude that stimulus evaluation and response preparation processes do not develop at the same rate. Although this approach was a step forward in estimating processes duration, it has several flaws [for criticisms, see 10, 11]. Foremost, the latency measures extracted from averaged ERPs are not necessarily valid. For example, the onset of the averaged ERP curve is unproportionally determined by the early occurrences of single-trial EEG deflections [10]. As a result, these latency measures are distorted and cannot be directly compared to RTs. This was well summarized by Callaway, Halliday, Naylor, Thouvenin [12]: "The latency of the average is not the average of the latencies".

More appropriate electrophysiological methods should allow researchers to identify specific processes measured by RT on every trial, instead of on an averaged signal, to get rid of such distortion. In the present study, electromyographic (EMG) activity was used to separate motor and non-motor components of individual RTs. The EMG activity can be recorded from the muscles primarily involved in the behavioural response. The excellent signal-to-noise ratio of EMG signals allows to decompose RT in two intervals separated by EMG onset on a trial-to-trial basis (Fig 1). Following Botwinick & Thomson [13], these two intervals are often termed premotor (PMT, from stimulus onset to EMG onset) and motor latencies (MT, from EMG onset to behavioural response). The functional interpretation of these two components deserves some comments. The motor time corresponds to time needed by the muscle to translate the cortico-spinal afferences into force and to reach the requested level of force to produce

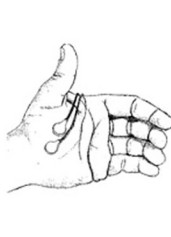
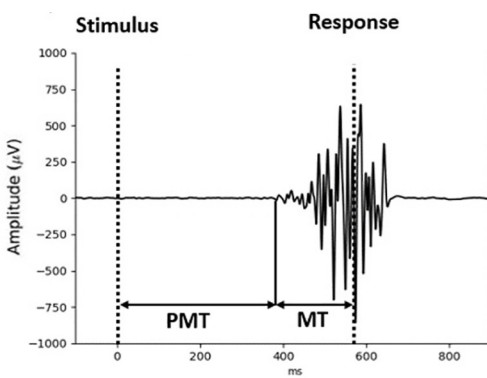

**Fig 1. Typical EMG signal.** Typical EMG signal is recorded from thumb muscles of 10-year-old child during performance of a task requiring response button presses (vertical dashed lines denote stimulus and response onsets; PMT denotes the premotor time and MT the motor time).

the overt response. As argued by Possamaï et al. [14] "If any portion of the reaction time solely reflect motor processes, surely it is motor time". Motor time, however, is not solely determined by low level, physiological mechanism. It largely depends on the rate of recruitment of the muscle's motor unit [15], and this rate largely depends on the synchronization of the afferent motor command [16]. As such, it reflects the quality of the central motor command, which depends on cognitive processes. As a matter of fact, several experimental manipulations have been shown to affect MT. For example, advance information about the effector to be used in the forthcoming response shortens MT [14]. Speed-Accuracy trade-off (SAT) also has a large impact on MT, with much reduced MT under hard speed pressure. SAT is largely considered as a prototypical **strategic** effect. The fact that MT is largely modulated by SAT indicates that it is also under strategic control. Perceptual manipulations can also affect this interval. Early work by Grayson [17] suggested that stimulus intensity could affect MT. More recently Servant et al. [18] and Weindel et al. [19] showed that color saturation and stimulus contrast, respectively, also affect MT. In contrast, several studies reported no effect of stimulus-response compatibility of MT [20]. To summarize, it appears that this interval clearly indexed motor processes, which can be modulated by cognitive factors, including strategic ones. Premotor time is more complex to specify functionally as it contains many different processes (mainly perceptual and decisional). Generally, it largely follows RT, as it is its main constituent. It is, however, important to note the PMT and MT are functionally independent, as they can be manipulated independently. Indeed, some experimental manipulations affect PMT while leaving MT constant (e.g., S-R compatibility [e.g., 20, 21], response bias [22]), while some other factor selectively impact MT, sparing PMT [e.g. 23]. Some factors even have opposite effects on the two intervals. For example, for erroneous responses, PMT is usually shorter, but MT is longer, compared to correct responses [24, 25]. Similarly, when force to press a response button is higher, MT is longer but PMT is shorter [20, 26].

Given this independence between the two intervals, fractioning RT based on EMG allows to efficiently assess the development rate of "motor" and "non-motor" components of RT.

The evolution of RT with age is very well described by a negative exponential function [1]:

$$RT = a + be^{(-c.age)} \tag{1}$$

where $a$ is the asymptote (the RT value obtained at adulthood), $a+b$ defines the intercept, and $c$ represents the decay rate (specifying the developmental rate of RT). To assess whether premotor and motor RT components develop non-linearly, this equation will be fitted separately to

PMT and MT. If both components are well described by this equation, the parameter *c* will provide estimates of the development rate of the different processes, allowing to directly compare them. In addition, the performance of children in each age group will be compared to that of adults to calculate the extent to which children are slower than adults. This analysis, called a Brinley plot [27], was traditionally performed on average RT data under various experimental conditions and involved plotting the mean RT of children against that of adults. Here, within-task Brinley plots will be calculated by plotting the average RT, PMT, and MT of children against those of adults. The common exponential decay rate for RT, PMT, and MT should result in a linear relationship between these variables and provide additional support for the hypothesis about the same developmental rate of RT components. Data collected as part of a larger project originally designed to study age-related changes in cognitive control [25] were used here to assess the developmental rate of PMT and MT. EMG data were acquired from a cross-sectional sample of children (N = 114) aged from 6 to 14 years old and a sample of 15 adults over 20 years of age (mean: 27 years old ± 4.5 years) who performed a child-friendly-version of the Simon task [28]. In this choice-RT task, the participants are instructed to press a left or a right button with the left or right thumb depending on the nature of an image (e.g. respond left for the banana). The image could be presented on the same side as the required response (compatible condition), or on the opposite side (incompatible condition). Cognitive control is more required in the incompatible condition to avoid pressing the ipsilateral response button, as indicated by longer RTs and a higher error rate [e.g. 20, 29]. Nonetheless, effects related to the development of cognitive control, which are not central for the present study, will be addressed in a separate publication. Independently from the involvement of cognitive control, EMG data recorded during performance of the Simon task are perfectly suitable to delimitate premotor and motor processes of RT. Both components were quantified, and exponential functions were applied separately to data in the compatible and incompatible task conditions.

## Method

### Participants

A cross-sectional sample of 148 children and 15 adults participated in the experiment. The data from thirty-four children were discarded from all analyses due to technical difficulties during EMG recordings (12 children), a failure to complete the task (4 children), or noisy EMG data (18 children). Details about the final cross-sectional sample are provided in Table 1. All children were recruited in the primary and secondary French schools. They provided verbal consent after being explained the experimental procedure and informed written consent was obtained from children's legal guardians and before the experiment. The group of adults consisted of university students and employees that were tested in the laboratory room at the university campus. They gave their informed written consent before the experiment. All participants had normal or corrected-to-normal color vision, and no history of neurological disorders. This work was approved by the head of the regional ethics committee (Comité de protection des Personnes, CPP, South-East, France).

### Material and apparatus

Children were tested individually in their schools during a single session of about 45 min. The experiment took place in a classroom adapted for experimental needs. Adults were tested in a dimmed laboratory room adapted for electrophysiological research. The experimental task was controlled by PsychoPy software [30]. All participants were seated at a table in front of the computer screen (a resolution of 1024 × 768 pixels and a refresh rate of 60Hz); the distance

**Table 1. Descriptive statistics.**

| Group (age in years) | Number of participants | Sex (number of boys) | Mean age (in months) | SD (in months) | Percent of rejected trials based on MAD |
|---|---|---|---|---|---|
| 6 | 14 | 8 | 81.1 | 4.0 | 3.5 |
| 7 | 12 | 7 | 93.3 | 3.3 | 3.5 |
| 8 | 15 | 8 | 103.7 | 3.0 | 2.8 |
| 9 | 13 | 7 | 116.6 | 4.3 | 2.3 |
| 10 | 11 | 6 | 129.2 | 3.6 | 2.1 |
| 11 | 12 | 6 | 138.1 | 4.0 | 3.0 |
| 12 | 12 | 6 | 150.3 | 3.1 | 2.4 |
| 13 | 12 | 6 | 164.3 | 3.6 | 2.2 |
| 14 | 13 | 6 | 173.3 | 3.5 | 2.3 |
| Adults | 15 | 7 men | 27 (years) | 4.5 (years) | 2.2 |

Characteristics of the participants. Number of trials rejected from statistical analysis.

between participants' eyes and the screen amounted to about 50 cm. Three sets of stimuli were used: cartoonish images of yellow banana and orange carrot (1st set), brown nut and red strawberry (2nd set), green frog and pink pig (3rd set). All stimuli (3.6˚ × 3.6˚) appeared in a black frame (12.1˚ × 3.9˚) presented in the center of the gray screen. Stimuli appeared at an eccentricity of 3.9˚ from the central fixation point (radius of diameter = 0.5˚). Responses to stimuli were provided via response buttons adapted to each individual child, mounted on cylindric handgrips placed on the table (Fig 2). Before the experiment, the height of the handgrip and of the thin cylinder (one of five handgrips and cylinders possible) were chosen independently to be adjusted to the size of participants' hands and thumbs (for adults the size of handgrips and cylinders were uniform). Participants were instructed to keep their thumbs on the response buttons during the entire task and to respond as quickly and as accurately as possible.

## Procedure

A child-friendly version of the Simon task [28] successfully used in the previous study [25], was administered to the participants (Fig 3). Each trial started with a fixation point that was displayed for 500 ms and followed by a centrally displayed black frame containing a target stimulus on its left or right side. The target was displayed on the screen until the response was given (no response time limit was applied) and 1 s later the next trial started. The participants

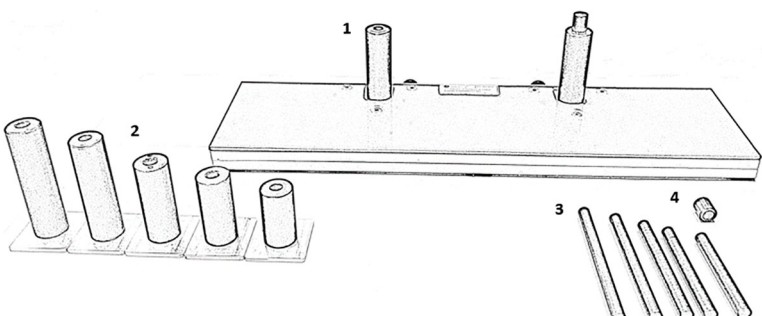

**Fig 2. Response device used in the experiment.** Response buttons were mounted on cylindrical handgrips (2) that contained another thin cylinder (3) touching force sensors located in the board (1). The sizes of the handgrip and the inner thin cylinder were chosen for each child to fit his/her hand size. The top of the thin cylinder was covered by a cap (4) to increase the pressing comfort.

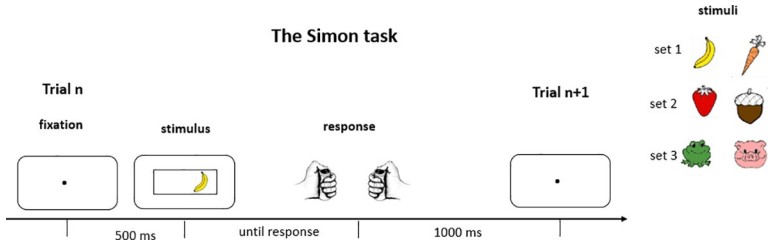

**Fig 3. The child-adapted version of the Simon task.** On the left: temporal sequence of events within one trial; on the right: three stimulus sets used in the task.

were instructed to press the left or the right button with their left or right thumb depending on the nature of the image. On compatible trials, the side of the required response matched the side of the target stimulus. On incompatible trials, the side of the required response mismatched the side of the target stimulus. Compatible and incompatible trials were equiprobable and presented in a random order. The assignment between images and response-sides was counterbalanced between participants within each age group. The experiment consisted of three blocks of 100 trials each, with short breaks every 25 trials and longer breaks between blocks. The experiment started with instructions providing the mapping between images and the required response side (this instruction was repeated at the beginning of each block). Next, two training blocks of 36 trials were administrated allowing the participants to learn the task. During the first training block, participants received an auditory feedback (two contrasting sounds for correct and incorrect response explicitly given before training). The next trial was initiated by the experimenter. In case a participant committed an error, the experimenter re-explained the task. During the second training block, no feedback was provided, and the inter-trial interval was always 1s. Each training block ended with a feedback on accuracy displayed on the screen and explained by the experimenter.

## Electrophysiological recording and processing

Ag/AgCl active flat electrodes (pre-amplified electrodes, Biosemi Inc., Amsterdam, The Netherlands) were used to record EMG activity of the flexor pollicis brevis of both hands (Fig 1). The electrodes were placed on the thenar eminence of each thumb with the maximal possible distance apart (1 cm at least). EMG activity was digitized online (sampling rate: 2048 Hz; analog bandwidth limit: –3 dB at 1/5th of the sampling rate) with use of the BioSemi Active-Two system (Biosemi Inc., Amsterdam, The Netherlands). The EMG signal was continuously monitored during the experimental task for the appearances of tonic muscular activity masking small task-related muscular activations. In such case, participants were asked to relax their hands.

Bipolar montages for left and right hand separately were computed offline and high-pass filtered at 10 Hz in order to remove slow fluctuations unrelated to EMG activity. Onsets of EMG activity were detected with a home-made custom program combining variance detection [31] and the "integrated profile" approach [32], written in Python. (This program is currently accessible upon request and soon will be released with open-source license). In a nutshell, after pre-processing to increase the signal/noise ratio (through the computation of the Teager-Kayser energy), to detect the presence of an EMG burst, the signal is compared to baselines values: when it exceeds the mean plus $n$ times the standard deviation ($n$ is a free parameter adjusted for each participant), an EMG burst is considered to have occurred. Its onset and offset are then estimated through its "integrated profile": the cumulative sum of the signal is calculated

from which the cumulative sum of the uniform distribution (i.e. a straight line) is suppressed. The onset is localised at the minimum of this difference and the offset at the maximum. Afterwards, EMG traces of each participant were inspected visually, and onsets of EMG bursts were corrected manually in case of inaccurate detection by the algorithm. When EMG bursts were accompanied by large tonic muscular activity making their onsets undetectable by visual inspection, they were excluded from the analysis (2% of all trials in each age group). During this procedure, the person who inspected the traces was unaware of the nature of the trial (compatible vs. incompatible), and of the age of the participant the traces corresponded to. Based on EMG onsets detection, trials including one EMG burst related to correct responses (78% per age group) were selected for further analysis, as they could be unambiguously decomposed into premotor time (PMT; from stimulus onset to the onset of EMG burst) and motor time (MT; from EMG onset to the button press defining the response) as illustrated in Fig 1.

## Data analysis

**Accuracy and chronometric indices.**   Standard analysis of variance was performed on accuracy and chronometric indices to allow comparison of results with other studies. Accuracy was calculated as the percentage of correct responses and analysis was performed on arsine transformed error rates. The chronometric indices were RT, premotor time (PMT) and motor time (MT). Outliers in the chronometric indices were identified and rejected based on median absolute deviation (MAD), a correction of central tendency and dispersion that is relatively insensitive to extreme values [33, 34]. For each participant and separately for the compatible and incompatible condition all RTs inferior and superior to median plus/minus 5 times MAD value were rejected from the analysis. Based on this criterion, about 2.6% of all trials were removed. The percentage of rejected trials did not depend on age ($F(9,119) = 1.6$, $p = .1$, $np2 = .1$, see Table 1). Jamovi 1.6.3, an open-source software for statistical calculations based on R programming language [35], was used for statistical analysis. The analysis of variance was performed on mean values of these indices calculated separately for each age group and for each experimental condition. Thus, ANOVA contained one within-subject factor Compatibility (compatible, incompatible) and one between-subject factor Age that had 10 levels (from 6 to 14 and adults).

**Exponential model fits.**   Fitting the negative exponential function (see Eq 1) to the mean data per age group was performed by minimizing the root-mean square deviation (RMSD) between the observed means and the predicted ones with a Simplex routine [36] as implemented in the SciPy Python toolbox [37]. Following Kail [1], the asymptote parameter $a$ was systematically fixed to the adults' performance value. This parameter was assessed in a separate analysis. Setting parameter $a$ to the adults' performance improved the quality of all fits as compared to leaving this parameter free to vary. For sake of brevity, this complementary analysis will not be presented here but is published on OSF page (see below). To reduce the risks of reaching local minima, one hundred independent iterations of the fitting procedure were performed, with different starting points drawn from uniform distributions around plausible parameter values. It was first evaluated whether mean RTs, mean PMTs, and mean MTs during performance on compatible and incompatible trials develop as a negative exponential function (see Eq 1). As it turned out to be the case, and since the decay rates of the exponential function (parameter $c$) looked very similar for all variables, it was then evaluated whether it is the same for all three variables and experimental conditions. For this purpose, we compared two models. The first model (*full* model) had six independent exponential functions fitted to RTs, PMTs and MTs on compatible and incompatible trials resulting in 12 free parameters (one scaling parameter $b$ and one decay rate parameter $c$ for each of the three dependent

variables in each compatibility condition). For the second model (*restricted* model), the decay parameter $c$ was constrained to be equal across dependent variables and conditions, resulting in 7 free parameters (one scaling parameter $b$ for each of the three dependent variables in each compatibility condition and one common decay rate parameter $c$ for all variables in the two conditions). Thus, the *restricted* model was nested in the *full* model. In such situation, the fit of the full model is necessarily better (or at least equal) to the fit of the *restricted* model since it is more flexible. Comparing the *full* and *restricted* models amounts to estimate whether the increase in the fit quality is large enough to compensate for the cost of additional number of parameters. This was assessed through the computation of the Akaike information criterion (AIC) [38], which provides a measure of predictive accuracy, defined as the ability to predict out-of-sample (i.e. new) data. The AIC was computed as follows [39]:

$$AIC = Nln\left(\frac{SS}{N}\right) + 2k \qquad (2)$$

where $N$ is the number of data points, $k$ is the number of free parameters plus one, and $SS$ is the sum of square of the vertical distances of each data point from the theoretical curve. The lower the AIC, the better the model. We also assessed whether the added parameters in the full model do bring statistically significant information through a likelihood ratio test whose values are distributed as a chi$^2$ distribution, with df being the difference in parameters between the full and restricted model (in the present case, df = 12–7 = 5). Finally, to asses the weights of evidence in favor of the selected model, we computed the Akaike weights (see Akaike [40], Burnham & Anderson [41], cited in Wagenmakers & Farnell [42]) and their ratio (see: Wagenmakers & Farnell [42]).

**Modified, within-task, brinley plots.**   While fitting Eq 1 to data provides essential information on the development rate, the smoothness of the fitted function does not allow to capture potential age-specific deviations, hence biasing towards a monotonic, non-linear, development. Another classical analysis used in developmental psychology is the Brinley plot approach [27] which displays the mean performance of children of a given age as a function of the mean performance of adults: It estimates, independently for each age group, one linear function that describes, for all variables together, the relation between children's and adult's values. This analysis thus makes it possible to determine how much slower the children are compared to adults. Such plots are normally assessed through different experimental tasks whose performance is obtained both in adults and in children [2]. While caution has been raised on the interpretation of the Brinley plot for developmental studies [43–45; see also 4 for discussions about the role of strategy and control processes for between-task differences], it remains an interesting statistical tool [46]. In the present context, modified Brinley plots can be built by plotting the different chronometric indices (RT, PMT and MT for the compatible and incompatible conditions) for each age group, as a function of their respective values in adults. Such "within-task" Brinley plots will be complementary to the fit above: while the previous exponential fits assess the development of each variable separately for all age groups together, the Brinley plots assesses for each age group whether all indices have a linear relationship, thereby suggesting a common decay rate compared to adults develop linearly when compared to adults.

**By-participant measures of proportion.**   The above analyses were done on means *per age group*. The developmental rates of PMT and MT might differ importantly between children of the same age, though, leading to artefactually create a relationship absent in individual data [47–49]. To evaluate the development of both premotor and motor components at the individual level, the proportion PMT and MT to RT was evaluated for all participants, including

adults. Thus, the ratio (mean PMT) / (mean RT) was computed to obtain the proportion of RT to which PMT corresponds. The same was done for MT (note that, by construction, the sum of these two ratios is necessarily equal to 1). As the ratio values turned out to be very similar across ages, the absence of the age effect was tested with Bayesian ANOVA performed with Jamovi 1.6.3: the likelihood of H0 (no age effect) vs H1 (age effect) was quantified separately for the compatible and incompatible task condition.

Data and scripts for all analysis can be found under the Open Science Foundation link of this project: https://osf.io/93rmd/?view_only=f0077de9ac33419cbe663cff6e4d413f.

## Results

### Behavioral and EMG results

An analysis of variance (ANOVA) on accuracy data with Compatibility and Age as within- and between-subjects factors respectively revealed a larger proportion of errors in the incompatible (5%) than the compatible condition (2%; $F(1,119) = 118.4$, $p < .001$, $n_p^2 = .5$), but no effect of age ($F(9,119) = 0.8$, n.s.). A second ANOVA conducted on mean RT showed a decrease of RT with age ($F(9,119) = 33.6$, $p < .001$, $n_p^2 = .72$). Mean RT were also longer on incompatible than compatible trials ($F(1,119) = 218.8$, $p < .001$, $n_p^2 = .65$), and the size of this compatibility effect decreased with increasing age ($F(9,119) = 3.7$, $p < .001$, $n_p^2 = .22$).

Another ANOVA conducted on PMT revealed a similar pattern of findings. PMT decreased with increasing age ($F(9,119) = 30.1$, $p < .001$, $n_p^2 = .70$) and was larger for incompatible than compatible trials ($F(1,119) = 254.3$, $p < .001$, $n_p^2 = .68$). The magnitude of this compatibility effect on PMT decreased with increasing age ($F(9,119) = 4.5$, $p < .001$, $n_p^2 = .26$).

A final ANOVA on the motor component of RT showed that MT also decreases with increasing age ($F(9,119) = 8.0$, $p < .001$, $n_p^2 = .38$). Consistent with previous EMG investigations of the Simon effect [e.g., 20, 50], no compatibility effect was observed on MT ($F(1,119) = 1.9$, $p = .2$), and compatibility did not interact with age ($F(9,119) = 1.4$, $p = .2$).

### Assessing the exponential decay rate of RT, PMT, and MT across development

Since both RT, PMT and MT decrease with age, it was next assessed whether the age effects on each of these variables follow the negative exponential function (Eq 1) by fitting this function to the development of each dependent variable in each compatibility condition separately (see Method for details). The fit quality of the *full* model is excellent, as reflected qualitatively by close observed and predicted values (Fig 4), and quantitatively by very high Pearson's product moment correlation coefficients (all $rs > .98$). All *c* values appear very close (Table 2), suggesting that RT, PMT and MT might develop at the very same rate.

To evaluate whether the data are better described by a single developmental decay rate, the *full* model was compared to a *restricted* model in which the decay rate parameter *c* was constrained to remain identical across dependent variables and compatibility conditions (see Method for details). Because the *restricted* model is nested in the *full* model, the fit quality of the former cannot be better than the later. However, the key question is whether the added complexity of the *full* model is worth given the improvement in fit quality. As Fig 4 shows, the fit of the *full* (gray solid lines) and *restricted* (dashed black lines) models are virtually identical. Being simpler, the *restricted* model (AIC = -499.991) outperformed the *full* model (AIC = -491.198). The superior performance of the *restricted* model is further outlined by the fact that the best-fitting decay rate estimates for the *full* model fall into a tight range (0.187 to 0.207) centered on the best-fitting estimate for the *restricted* model (0.198). It is also supported by a likelihood ratio test that revealed that the full

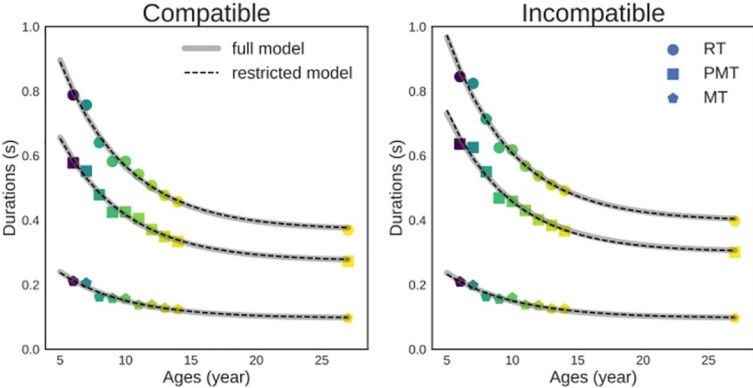

**Fig 4. Observed data and best-fitting *full* and *restricted* model.** Observed data (circles for RT, squares for PMT and pentagons for MT) versus predicted data from the best-fitting *full* model (grey solid lines) and the best-fitting *restricted* model (black dashed lines). For models, the asymptote parameter *a* of the exponential function was set to the adults' performance value. For the *full* model, the two remaining parameters of the exponential function (scaling *b* and decay rate *c*) varied independently for each of the three dependent variables (RT, PMT and MT) in each compatibility condition. The *restricted* model was similar to the *full* model except that the decay rate parameter *c* was constrained to be identical across variables and conditions.

model does not add any significant information compared to the restricted one ($Chi^2_5 = 0.039$, $p = 0.99998$). The ratio of the relative AIC weights (see Wagenmakers & Farnell [42]) indicates that the restricted model is 82 times more likely than the full one.

Similar comparison between the *full* and *restricted* models, was performed 1) on seven data sets: RT, PMT and MT compatible and incompatible (as in above described case) and additionally on the difference calculated by subtracting PMT compatible from PMT incompatible (as suggested by one reviewer, such subtraction would allow to separate control processes involved in task performance on incompatible trials and comparing their decay rate to MT would provide an additional and strong support for common maturation rate of non-motor and motor processes); and on 2) individual data points instead of on means per age group. Both additional analyses have shown that *restricted* models had smaller AIC value (-594.51 in case of first and -4587.7 in case of the second analysis) than *full* models (AIC = -577.32 for the first analysis and AIC = -4076.1 for the second one). Thus, a common decay rate would describe the maturation of all variables and also when the calculation is based on individual data points. We do not report in detail these analyses (they can be found under the Open Science Foundation link of this project), although they both support our conclusions since the first relies on a strong assumption of pure insertion criticized in the introduction, and because the second does not allow to restrict the model by fixing the parameter *a* (asymptote) to adult's performance.

Altogether, these findings provide strong support for a common decay rate that describes the processing speed evolution of all dependent variables.

**Table 2. Best-fitting parameter estimates for the full exponential model and the restricted exponential model.**

|  | RT | | | | PMT | | | | MT | | | |
|---|---|---|---|---|---|---|---|---|---|---|---|---|
|  | Comp | | Incomp | | Comp | | Incomp | | Comp | | Incomp | |
|  | *b* | *c* | *b* | *c* | *b* | *c* | *b* | *c* | *b* | *c* | *b* | *c* |
| Full | 1.377 | 0.194 | 1.582 | 0.202 | 1.009 | 0.195 | 1.239 | 0.207 | 0.367 | 0.192 | 0.345 | 0.187 |
| Restr | 1.423 | 0.198 | 1.538 | 0.198 | 1.036 | 0.198 | 1.159 | 0.198 | 0.387 | 0.198 | 0.378 | 0.198 |

Full: full model; Restr: restricted model; Comp: compatible condition; Incomp: incompatible condition; b and c: scaling and decay rate parameters of the exponential function.

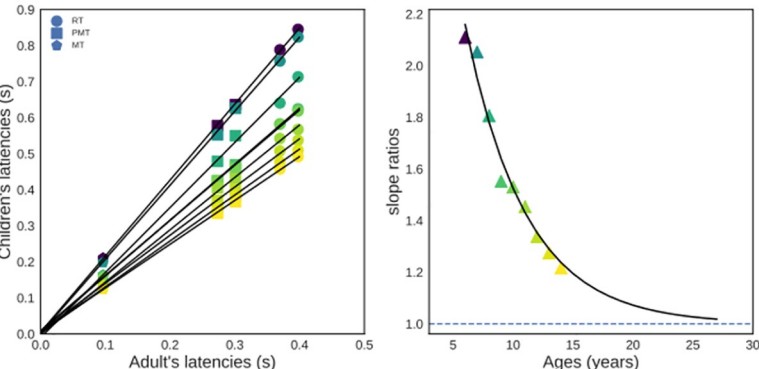

**Fig 5. Brinley plots.** Left panel: within-task Brinley plots. Each plot displays mean RT (circles), PMT (squares) and MT (pentagons) for each compatibility condition from children in a given age group (color code gradually changing from dark violet for 6-years-olds to yellow for 14-years-olds) against adult's performance (x-axis). Black lines show the best fit of linear function capturing the relation between the six data point for a given age. Slope value of each linear function is plotted in the right panel. Right panel: slope values (triangles) from linear fits (y-axis) as a function of age (x-axis). The black line represents predictions from the best-fitting exponential model. The decay rate parameter (*c*) of this model was set to that previously obtained from the restricted model, and the asymptote (*a*) was set to 1, leaving only one free parameter (scaling *b*) to capture the data.

## Modified, within-task, Brinley plots

The above analyses were conducted across all age groups, which may have smoothed subtle age differences. To address this concern, we compared children's mean performance in each condition against adults' mean performance. Such Brinley plots are usually conducted on mean RT data across various experimental tasks. Here, the six datapoints for each age group (mean RT, mean PMT and mean MT on compatible and incompatible trials) were used to construct within-task Brinley plots. Note that this analysis is somehow orthogonal to the previous one, since, while fitting Eq 1 was done *for each variable across all age* groups, the present analysis will be performed *for each age group, across all variables*. The common exponential decay rate for RT, PMT and RT in the two task conditions should translate into a linear relationship with a slope greater than 1 and an intercept of 0 for a given age [1]. The results of this analysis are depicted in Fig 5 (left panel). For each age group, the data are very well described by a linear relationship (all $R^2 > .998$), the intercept of which being systematically close to 0 (range -0.01 to 0.006) and the slope being superior to 1 (range 1.22 to 2.11). If RT, PMT and MT speed-up at the very same rate, a second prediction is that the slope values from linear regressions should decrease exponentially as a function of age (Eq 1), with a decay rate equal to the one extracted from the restricted model and an asymptotic value equal to 1 [1]. To test this prediction, we fit Eq 1 to slope values from our linear regressions, setting the decay rate parameter *c* to 0.198 (best-fitting parameter from the restricted model, see Table 2) and the asymptote parameter *a* to 1, leaving only one free parameter (scaling *b*) to capture the data. Despite these stringent constraints, the fit quality of the model is excellent ($r = .987$), with a near perfect match between observed and predicted data (Fig 5, right panel). These findings provide strong converging evidence for a common developmental rate between RT, PMT, and MT.

## Proportion of RT accounted for by PMT and MT: analysis of individual data

The above analyses (similar decay rate and linear Brinley plots) strongly suggest that the PMT and MT evolve proportionally during childhood: both components speed-up in duration with

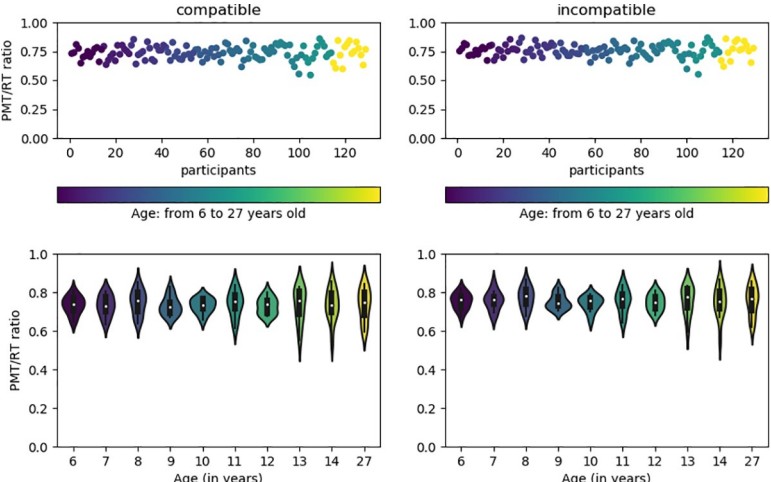

**Fig 6. The ratio between PMT and RT calculated for individual participants.** Top panel: points are means calculated across all trials separately for each participant in the compatible (on the left) and the incompatible (on the right) task condition. Value 1 on the y axis represents normalized duration of RT, point values represent the proportion of PMT within this normalized space. Colors of points follow the participant's age: dark velvet for youngest children (6 years old) and yellow for adults. Bottom panel: probability density of the data at for different age groups. White marker represents median of the data, box indicates the interquartile range.

the same rate. However, such result was obtained on data averaged across all children of the same age. To directly assess whether PMT and MT always correspond to the same proportion of RT across all ages, the ratio between PMT and RT was calculated for each participant, including adults, separately for the compatible and incompatible task condition. The data were normalized so that all mean RT were set to be equal to 1, and the position of the mean PMT was plotted in this normalized space (Fig 6). Independently of age, the ratio values are very similar across participants, without any apparent developmental trend. In order to quantify the evidence in favor of the absence of age effect on ratio values, Bayesian hypothesis testing was used. Specifically, Bayesian ANOVA with default priors was applied to test the likelihood of H0 (no age affect) vs H1 (age effect). Following Jeffrey's heuristic rule [51], Bayesian Factor (BF) between 1 and 3 was considered as anecdotal evidence, between 3 and 10 as moderate evidence, between 10 and 30 as strong evidence, and larger than 150 as very strong evidence. For sake of simplicity, the Bayes Factor BF01 was computed separately for compatible and incompatible conditions. As already shown by frequentist analysis of ANOVA, the difference between compatible and incompatible conditions was present for PMT, but not for MT. Therefore, by construction of the ratio value, ratios for the two task conditions also differ. The results of the Bayesian ANOVA indicate that for both conditions there was strong evidence for no effect of age (BF01 = 38.7 and 27.8 for compatible and incompatible, respectively). It means that the relative developmental rate in duration of PMT and MT remains stable across childhood, up to adulthood.

## Discussion

Cognitive abilities show a massive non-linear development through childhood and adolescence, well characterized by the evolution of response times (RT). The previous attempts to describe developmental dynamics of the different processes included in the RT suffer from methodological bias or unwarranted assumptions. In the current study, based on the EMG activity of response effectors, RTs were fractionated into a premotor time (PMT; from stimulus

presentation to EMG onset) and a motor time (MT; from EMG onset to the behavioral response) on a trial-by-trial basis, providing an unequivocal estimation of the duration of two RT components. Since PMT occupies most of the RT duration, it is not surprising that it decreased non-linearly across ages in the present study. However, further work is needed to delineate processes encompassed by PMT and to describe their developmental pattern. This important objective could be achieved by adopting the computational modeling approach allowing to decompose PMT into subcomponents. More surprising is the result strongly indicating that MT also decreased across ages in a non-linear way (see below for detailed discussion). Overall, current results suggest that various processes encompassed by the RT develop rapidly in early childhood, gradually slowing down in late childhood and reaching a level of stability in early adulthood. Furthermore, fitting the data to a negative exponential function allowed to extract a parameter describing the speed at which the function decreases towards asymptote and thus to directly compare developmental rates of PMT and MT. This comparison clearly showed that PMT and MT durations decrease at the very same rate across childhood and adolescence. This was confirmed by further analyses, including that conducted at the level of individual participants. Therefore, the current results indicate that PMT and MT, two RT components reflecting different set of processes, have the same developmental trajectory.

The very same functional development rate across ages for premotor and motor processes is reminiscent of the so-called global maturation model underpinned by a general increase in information processing speed [1, 2, 52, 53]. This proposition emerged from the observation that a similar decay parameter $c$ of the exponential function (see Eq 1) fits developmental data from a large variety of cognitive and perceptual-motor tasks, pointing to a common maturation factor. It is to be noted that while the global maturation model predicts that all processes within a task should mature at the same speed, most of evidence has been based so far on between tasks comparisons [1, 2, 54]. The present data provide the first unbiased test of the hypothesis that all processes develop at the same speed, and, somehow surprisingly, the same maturation rate of premotor and motor components strongly support a global maturation of the different processes, as assessed by their processing speed.

Such a global maturation view has received several criticisms. First, the strategy for executing even simple cognitive tasks (i.e. the processes involved and their organization) may vary across the lifespan, precluding a direct chronometric comparison [4]. Second, the regression approach of Brinley plots was criticized for concealing task-specific age-related differences by providing only a single regression equation across tasks and conditions that might be erroneously considered as supporting a global mechanism view [9, 55]. Consequently, researchers shifted their focus on the maturation of specific cognitive processes and qualitative developmental changes. Such domain-specific approach dovetailed with results from developmental neuroscience, showing important regional variations in brain maturation [56–59]. Both in vivo neuroimaging [60–64] and post-mortem [65–67] studies have consistently reported earlier development of brain gray and white tissue in core sensory and motor regions, and later development in frontal and temporal regions. Since PMT duration predominantly depends on association cortices, it could be expected that the maturation in processing speed of the premotor component will be slower than that of the motor component, which depends predominantly on motor brain regions.

The maturity of the brain is most often studied at two levels independently. One level is related to the functional maturity, that is to the efficiency of cognitive processes at fulfilling their function. The second level is related to physiological maturation, that is to the structure of the brain in terms of white and grey matter characteristics. At minimum, the present data contrast with brain maturation data if a causal link between physiological and functional

maturation is assumed. Therefore, any integrative theoretical proposition requires solving the apparent paradox between the protracted development of motor component, as shown by the current results, and the rather early maturation of motor brain areas, as shown by gray and white matter maturation trajectories. In other words, the present results prompt developmental sciences to reconcile global and domain-specific conceptions of cognitive maturation. Reaching such a goal requires to question a fundamental, more or less implicitly made assumption of a causal link between physiological maturation and functional efficiency. Beyond the developmental aspect, it is a common observation that the functional efficiency varies largely depending on context, for example when a person is distracted by noise or an unexpected event. Such changes in efficiency may occur on a very short time scale, which obviously do not invoke alterations in brain structure (and even more when one considers the reversibility of those context effects). Extending this argument on current results, the link between efficiency of various RT processes and brain maturity might not be as straightforward as it may seem. While a physiologically immature system is very unlikely to be functionally fully efficient, the reverse is not necessarily true: the fact that a system has reached physiological maturation does not imply that it is always functionally fully efficient. Both brain maturation and behavioral data in simple reaching tasks [68] lead to expect the motor system to be efficient at early ages already. Assuming it is indeed the case, the protracted developmental trajectory of the motor component of RT, as observed in the current study, requires speculative explanation. A possibility would be that the motor component is functionally held back in its speed to be adjusted to the speed of processes taking place within the premotor RT component. This hypothetical adjustment might be an adaptive developmental mechanism ensuring optimal flow of information within cognitive modules and underlying brain regions. Large differences in processing speed between different operations might actually be *harmful* for task performance, as less mature processes could be saturated by excessive flow of information from more mature processes. By slowing down more mature processes to reach the processing speed of the slowest one, such mechanism would mimic a global maturation pattern. This hypothesis predicts that the developmental rate of MT can vary between tasks, depending on the speed of the slowest process involved. It may seem that this prediction can readily be tested by comparing the two experimental conditions (compatible and incompatible) of the present dataset, since they differ in PMT. This is, however, not the case because the two conditions were randomly administrated within one experimental block and thus the nature of the upcoming trial could not be predicted in advance. If our proposed speed optimization mechanism is valid, it is probable that it would operate globally, being set once for the entire task. This prediction should be tested in future work. It is possible that the speed optimization mechanism is not only developmental but could instead be implemented whenever certain factors (e.g. pathologies, specific task conditions) imply some differences in the speed of cognitive operations.

Pointing to possible limitations of the current study, its cross-sectional nature may conceal more subtle age-related effects in the maturation of the duration of RTs components. This must be considered especially as the final size of the age groups is rather modest and only chronological age was considered. Thus, important individual variations in processes maturation was not taken into account. Finally, the method of separating PMT and MT is ubiquitous, but the functional interpretation of these RT components is not. Whereas MT can be considered as reliably estimating motor component, the influence of motor processes on PMT cannot be ruled out.

Despite these limitations, the current data strongly indicate that the premotor and motor components of RT develop with the same rate throughout childhood and adolescence. Given the data on brain development, this result suggests a decoupling between structural maturity

and functional efficiency that could be adjusted to the efficiency and processing speed of other processes involved in task performance.

## Acknowledgments

The authors wish to thank D. Louber and D. Paleressompoulle for designing and setting-up the response device.

## Author Contributions

**Conceptualization:** Kamila Śmigasiewicz, Mathieu Servant, Agnès Blaye, Borís Burle.

**Data curation:** Kamila Śmigasiewicz, Borís Burle.

**Formal analysis:** Kamila Śmigasiewicz, Mathieu Servant, Borís Burle.

**Funding acquisition:** Agnès Blaye, Borís Burle.

**Investigation:** Solène Ambrosi.

**Methodology:** Kamila Śmigasiewicz, Mathieu Servant, Borís Burle.

**Project administration:** Agnès Blaye, Borís Burle.

**Resources:** Agnès Blaye.

**Software:** Kamila Śmigasiewicz, Solène Ambrosi, Borís Burle.

**Supervision:** Agnès Blaye, Borís Burle.

**Visualization:** Kamila Śmigasiewicz.

**Writing – original draft:** Kamila Śmigasiewicz.

**Writing – review & editing:** Mathieu Servant, Agnès Blaye, Borís Burle.

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
