## [Decision Letter · Decision Letter 0]

24 Feb 2021

PONE-D-21-00194

Speeding-up while growing-up: synchronous functional development of motor and non-motor processes across childhood and adolescence

PLOS ONE

Dear Dr. Śmigasiewicz,

Thank you for submitting your manuscript to PLOS ONE. After careful consideration, we feel that it has merit but does not fully meet PLOS ONE’s publication criteria as it currently stands. Therefore, we invite you to submit a revised version of the manuscript that addresses the points raised during the review process. It is expected that you respond to all comments from the 3 reviewers and update the manuscript accordingly. Of note, I agree with the final comment of Reviewer 3; please refrain from putting so much emphasis on the development of brain structure in the abstract, as this was not assessed in the current research. 

We look forward to receiving your revised manuscript.

Kind regards,

Bradley R. King

Academic Editor

PLOS ONE

Journal Requirements:

2. We noted in your submission details that a portion of your manuscript may have been presented or published elsewhere.

"The present study is based on a previously collected dataset. Former analyses were focused on erroneous responses and their EMG burst correlates (Śmigasiewicz, K., Ambrosi, S., Blaye, A., & Burle, B., 2020. Inhibiting errors while they are produced: direct evidence for error monitoring and inhibitory control in children. Developmental Cognitive Neuroscience, 41, 100742). In contrast, the present study is based uniquely on correct responses and aims at fractioning RTs. Thus, not only the theoretical question asked in the current study, but also the methodological approach are different in the two studies. Furthermore, to trace the evolution of processing speed to adulthood, we have completed the previous data set by including a group of young adults."

Please clarify whether this publication was peer-reviewed and formally published. If this work was previously peer-reviewed and published, in the cover letter please provide the reason that this work does not constitute dual publication and should be included in the current manuscript.

Reviewers' comments:

Reviewer's Responses to Questions

**Comments to the Author**

1. Is the manuscript technically sound, and do the data support the conclusions?

Reviewer #1: Partly

Reviewer #2: Yes

Reviewer #3: Partly

2. Has the statistical analysis been performed appropriately and rigorously? 

Reviewer #1: I Don't Know

Reviewer #2: Yes

Reviewer #3: Yes

3. Have the authors made all data underlying the findings in their manuscript fully available?

Reviewer #1: Yes

Reviewer #2: Yes

Reviewer #3: Yes

4. Is the manuscript presented in an intelligible fashion and written in standard English?

Reviewer #1: Yes

Reviewer #2: Yes

Reviewer #3: Yes

5. Review Comments to the Author

Reviewer #1: In this manuscript, the authors report the results of a study on the development of information processing speed during childhood and adolescence. A common Simon task was used, in which EMG was measured. Based on the EMG signal “premotor time” and “motor time” could be studied separately. The change in PMT and MT could be described by a negative exponential function, which indicated a similar development trajectory for both parts of the process. The researchers argue that this could indicate a common maturation factor. This idea is consistent with the global maturation theory, but contrasts with the finding that different brain regions mature at different rates. They also speculate that the common decay could be the result of a speed optimization mechanism whereby the faster processes are adapted to the speed of the slower processes.

This is an interesting and relevant work for researchers in developmental psychology. The article is well-written, the method seems solid (but see major comment below) and the results are clear.

I have only one general and a few smaller comments.

The study relies heavily on the assumption that PMT and MT can be separated on the basis of EMG.

1. Can the authors provide the necessary evidence for this? What is the validity of this method? What is premotor and motor here? These terms should be clearly operationalised.

2. Related to this, in the discussion PMT and MT are linked to association cortices and motor brain regions, respectively. Since EMG measures polarization at the periphery, it is unclear to me to what extent you can make that link between EMG timing and brain processes.

If the authors can clarify this method / concept, this will facilitate interpretation of the results and discussion. If necessary, the limitations of this method or the associated assumptions can be acknowledge as a limitation at the end of the discussion.

Minor points:

L49: Throughout the paper you refer to “cognitive function”. Whether you call reaction time and the underlying processes “cognitive” or “motor” may depend on your background. I would recommend adding a short footnote at the beginning of the paper to clarify what you mean by “cognitive”.

L59, typos: a non-linear evolution characterizeS the maturation…., If so, WHETHER they have

L61-62: Please elaborate or provide a number of examples to clarify what you mean with tasks and tasks conditions here.

L64, typo: changes of RT WERE

L71: Please explain what you mean with “insertion of processes” and the “insertion hypothesis”

L71, typo: do you mean “the quantity OR nature of information”?

L81: write LRP in full please

L106-107: This statement requires scientific evidence / support. See major comment.

L142-143: The sample eventually consisted of 116 children, across a rather wide age range. This is considered to be small for behavioural research. I appreciate that the effects in your study are large, but I would ask to address the size of the sample briefly.

L190: Is my understanding correct that the time between the appearance of the fixation point and the stimulus was fixed? I can imagine that such a fixed period of time can give rise to anticipation. Have you checked for this? How did you deal with this?

L241: I admit that I am not a specialist in this matter, but in my opinion the Data Analysis section is clear and very well written.

L266: Is it possible to include this in the OSF page?

L295, typo ?: While fitting eq 1 to THE data provideS essential information ..

L301, typo:… all variableS…

L321, typo: ... to which PMT correspondS ...

L367, Table 2, typo: First parameter b in lowercase

L439: bayes factor previously written as Bayesian Factor. Remain consistent.

L486: Clarify what you mean with tasks comparisons or provide an example (see also L61-62)

L519: Efficiency of various RT processes and brain maturity. At first, I couldn't fully follow, but later it became clear to me that this referred to brain function vs. brain structure. It seems useful to me to explain this briefly.

L544: Within or before your concluding paragraph, a short section with the limitations of your work (e.g. assumptions related to the separation of PMT / MT based on EMG, sample size) is required. Here, it is important to note that your conclusions related to development are inferred from cross-sectional data. In that context it should be noted that you talk about biological processes (structural maturation and functional efficiency) while you have classified the groups based on chronological age. We know that there is great individual variation in timing and rate of maturation. These are minor points, which need to be acknowledged nonetheless.

Reviewer #2: The paper “Speeding-up while growing up: synchronous functional development of motor and non-motor processes across childhood and adolescence" examines changes in reaction time on a sample of children 6-14 years old. Using EMG, they fractionate the RT into pre-motor and motor components and show that the changes in both components develop at the same rate.

Overall, the paper is well done and I believe it makes a useful contribution to the literature. I have a couple of major comments and some other minor comments for improvement.

Major comments:

1. In the introduction, it would be helpful to explain what the theoretical bases of the premotor vs. motor components of RT are from prior work. For example, several studies show that many changes in RT are almost fully correlated with the premotor component, and very little with the motor component. Are there studies that show changes in the motor component of RT and if so, what would be the basis for such changes?

2. Given the research question, the averaging of all participants within an age-group throws away a lot of information. In addition, this also reduces the number of data points that need to be fit by the exponential, which explains the high R^2 values. The authors should provide more individual level dot plots (rather than just the group averages) and also report the robustness of this curve fitting to removing some of the data points (to avoid any effects of a few outliers).

3. The last part of the Discussion about “slowing down” processes to ensure “optimal flow of information” seems to be extremely speculative. Are there any known mechanisms that could affect the motor component of RT in a task-specific manner? The authors should also consider if there are other simpler alternatives for changes in the motor component of RT (e.g., the vigor with which participants respond, which can potentially shorten the time between the EMG burst and the response). Also, including this extremely speculative part in the abstract also makes it sound like a “likely” explanation when there is very little evidence for this (at least at the moment).

Minor comments:

1. Ln 99 – there seems to be a missing reference

2. Ln 226 – the EMG onset detection algorithm could be explained in a bit more detail here so that the manuscript is more self-contained

3. At several points in the manuscript, the authors refer to information that is available upon request – it might be easier to add this information as supplementary material (especially given that some of the material is already on OSF)

4. Ln 248 – insensitive (instead of unsensitive)

5. The description of the Brinley plots could be motivated earlier (presenting the rationale for the analysis in the results section makes it much more difficult to anticipate for the reader)

Reviewer #3: In this paper, the authors decomposed response time into reaction time (PMT: time interval between stimulus onset to EMG onset) and movement time (MT: time interval between EMG onset and behavioral response) in a modified Simon task and found that both PMT and MT follow the same developmental trajectory from 6 years of age to adulthood. This paper is well-written, methods are clearly described, and analyses are appropriate. I am sure that the results themselves will contribute to the literature of developmental science. However, I have a major concern regarding the conclusion inferred from the data and I strongly suggest the authors performing one additional piece of analysis to address this concern. I describe it below:

I believe that the major argument the authors attempted to make is that behavioral signatures of cognitive (PMT) and motor processes (MT) follow the same developmental trajectory, contradicting the fact that the physiological development landscapes between cognitive and motor related brain areas are distinct. MT largely reflects a motor process, my concern, however, is that PMT does not reflect non-motor processes only. Clearly, besides perception (which is actually trivial in compatible trials as perception in these cases is primarily driven by automatic processing), premotor processes include action selection, and movement preparation, etc., which are generally thought of as motor processing, and related to premotor cortex (see Wong et al. 2015 Motor planning). However, the neural evidence of non-motor process (i.e., cognitive control) in the discussion section is mostly about temporal and frontal cortex. Thus, the behavioral measurements demonstrated in this paper and the existing evidence of neural development are not an apples-to-apples comparison. Some experiments where PMT is dominated by cognitive processes could provide a better comparison. One candidate is go/nogo task or stop-signal task that requires only one response. In these cases, the pre-motor interval is largely about cognitive control and may not be contaminated by motor-related process such as action selection. However, I do NOT suggest any additional experiments. The other way to address my concern is to perform an additional piece of analysis on the existing dataset. Unlike PMT in compatible trials, PMT in incompatible trials consists of an additional cognitive process (i.e., response inhibition which is related to frontal cortex) that addresses the competition between automatic processing induced by the spatial location of stimulus and the computational processing induced by the identity of the stimulus. This cognitive control function could be inferred by the PMT difference between these two types of trials. The result that this PMT difference follows the same nonlinear developmental trajectory with MT would strongly support the authors’ argument. Otherwise, the current conclusion would be undermined.

Another comment:

I found the abstract is a bit misleading as half of it is talking about neural development, but there is no neural data at all in this paper. I would suggest to focus on the data and results instead but I will leave the decision to the authors.

6. PLOS authors have the option to publish the peer review history of their article (what does this mean?). If published, this will include your full peer review and any attached files.

Reviewer #1: **Yes: **Frederik Deconinck

Reviewer #2: **Yes: **MEI-HUA LEE

Reviewer #3: No

---

## [Author Response · Author response to Decision Letter 0]

13 May 2021

Ad 2

The present study is based on a large dataset collected in children that included both behavioural and psychophysiological measures. Subset of these data was published in peer-review journal Developmental Cognitive Neuroscience (Śmigasiewicz, K., Ambrosi, S., Blaye, A., & Burle, B., 2020. Inhibiting errors while they are produced: direct evidence for error monitoring and inhibitory control in children. Developmental Cognitive Neuroscience, 41, 100742). These published analyses were focused on the electromyographic activity (EMG bursts) of erroneous responses to answer the question how errors monitoring processes develop across childhood. The present study does not constitute in any ways a dual publication because the theoretical question asked in the two manuscripts are dramatically different, as well as the methodological approach applied to address these questions. First, in the published manuscript, we focussed on EMG activity on erroneous trials, while errors are not analysed in the current manuscript. Second, we here do not analyse EMG activity for itself, but rather we use it to decompose behavioural measure (reaction times of correct responses) into underlying components. Furthermore, to trace the evolution of processing speed to adulthood, we have completed the previous data set by including a group of young adults. The Open Science movement, that your journal support, encourages sharing data for re-analyses. We agree with this view and believe that it is in the best interest of our society, to (re)use the data already collected to answers new scientific questions. It is in this logic that we reprocessed, for an entirely different purpose, data that we already published elsewhere. Obviously, the raw data will be available on the Open Science Foundation website.

Responses to each reviewer's comments are uploaded in a separate file.

---

## [Decision Letter · Decision Letter 1]

18 Jun 2021

PONE-D-21-00194R1

Speeding-up while growing-up: synchronous functional development of motor and non-motor processes across childhood and adolescence

PLOS ONE

Dear Dr. Śmigasiewicz,

Thank you for submitting your manuscript to PLOS ONE. After careful consideration, the reviewers were largely satisfied with the updates to the manuscript. Reviewer 1 pointed out a few typos that would be worth fixing. And, more critically, Reviewer 3 made some final suggestions in order to improve the statistical analyses. Accordingly, we invite you to submit a revised version of the manuscript that addresses the remaining points raised during the review process. It is worth noting that this is considered a minor revision. Accordingly, the revised manuscript will NOT be sent back to reviewers. Rather, I will check the updates. This will ensure a faster review time. 

We look forward to receiving your revised manuscript.

Kind regards,

Bradley R. King

Academic Editor

PLOS ONE

Journal Requirements:

Reviewers' comments:

Reviewer's Responses to Questions

**Comments to the Author**

1. If the authors have adequately addressed your comments raised in a previous round of review and you feel that this manuscript is now acceptable for publication, you may indicate that here to bypass the “Comments to the Author” section, enter your conflict of interest statement in the “Confidential to Editor” section, and submit your "Accept" recommendation.

Reviewer #1: All comments have been addressed

Reviewer #3: All comments have been addressed

2. Is the manuscript technically sound, and do the data support the conclusions?

Reviewer #1: Yes

Reviewer #3: Yes

3. Has the statistical analysis been performed appropriately and rigorously? 

Reviewer #1: Yes

Reviewer #3: Yes

4. Have the authors made all data underlying the findings in their manuscript fully available?

Reviewer #1: (No Response)

Reviewer #3: Yes

5. Is the manuscript presented in an intelligible fashion and written in standard English?

Reviewer #1: Yes

Reviewer #3: Yes

6. Review Comments to the Author

Reviewer #1: All my comments have been addressed. I only spotted a few typos:

L55: of A series of

L64: StRoop task

Reviewer #3: Thanks for your effort to address my previous comments. There are two suggestions to your analyses:

1. The error data (in percentage) seems not follow the normal distribution assumption for ANOVA (I assume that most participants have very low error rates and they are bounded between 0 and 1). You may at least provide the test for the normal distribution assumption or use some alternatives such Poisson or Beta regression depending on the distributions the data follow.

2. Since two exponential models used to fit the data are nested, the likelihood ratio test would give a formal statistical comparison between models, rather than an arbitrary comparison given by AIC. That is, you could statistically reject the hypothesis that the full model is better than the restricted one.

I believe that these two suggestions will not affect the significance of your study, but it would improve the validity of the statistical analyses.

7. PLOS authors have the option to publish the peer review history of their article (what does this mean?). If published, this will include your full peer review and any attached files.

Reviewer #1: **Yes: **Frederik Deconinck

Reviewer #3: No

---

## [Author Response · Author response to Decision Letter 1]

24 Jul 2021

Reviewer #1: All my comments have been addressed. I only spotted a few typos:

L55: of A series of

L64: StRoop task

Response: Done, thank you for careful reading.

Reviewer #3: Thanks for your effort to address my previous comments. There are two suggestions to your analyses:

1. The error data (in percentage) seems not follow the normal distribution assumption for ANOVA (I assume that most participants have very low error rates and they are bounded between 0 and 1). You may at least provide the test for the normal distribution assumption or use some alternatives such Poisson or Beta regression depending on the distributions the data follow.

Response: In response to this comment, we now present in paper ANOVA on arsine transformed error rates. ANOVA performed on these new values confirmed the previous analysis, therefore in method section we just added that analysis was performed on arsine transformed error rates and we adjusted F and p values in the result section.

2. Since two exponential models used to fit the data are nested, the likelihood ratio test would give a formal statistical comparison between models, rather than an arbitrary comparison given by AIC. That is, you could statistically reject the hypothesis that the full model is better than the restricted one.

I believe that these two suggestions will not affect the significance of your study, but it would improve the validity of the statistical analyses.

Response: 

We thanks the reviewers for pointing out this possibility. We have indeed added the likelihood ration test, which clearly confirms that the full model does not add any significant information to the fit. To get an idea of the weight of evidence in favor of the two model, we also computed the weighted AIC (see Wagenmakers & Farnell, 2004) which revealed that the restricted model is 82 more likely than the full one, given the data.

---

## [Editor Report · Decision Letter 2]

27 Jul 2021

Speeding-up while growing-up: synchronous functional development of motor and non-motor processes across childhood and adolescence

PONE-D-21-00194R2

Dear Dr. Śmigasiewicz,

We’re pleased to inform you that your manuscript has been judged scientifically suitable for publication and will be formally accepted for publication once it meets all outstanding technical requirements.

Kind regards,

Bradley R. King

Academic Editor

PLOS ONE
---

## [Editor Report · Acceptance letter]

31 Aug 2021

PONE-D-21-00194R2 

Speeding-up while growing-up: synchronous functional development of motor and non-motor processes across childhood and adolescence. 

Dear Dr. Śmigasiewicz:

I'm pleased to inform you that your manuscript has been deemed suitable for publication in PLOS ONE. Congratulations! Your manuscript is now with our production department. 

Kind regards, 

on behalf of

Dr. Bradley R. King 

Academic Editor

PLOS ONE